# What predicts response to sertraline for people with depression in primary care? a secondary data analysis of moderators in the PANDA trial

**Charlotte Archer**[1]*, **David Kessler**[1], **Gemma Lewis**[2], **Ricardo Araya**[3], **Larisa Duffy**[2], **Simon Gilbody**[4,5], **Glyn Lewis**[2], **Tony Kendrick**[6], **Tim J. Peters**[7], **Nicola Wiles**[1]

1 Population Health Sciences, Bristol Medical School, University of Bristol, Bristol, United Kingdom, 2 Division of Psychiatry, University College London, London, United Kingdom, 3 Health Services and Population Research Department, King's College London, London, United Kingdom, 4 Department of Health Sciences, University of York, York, United Kingdom, 5 Hull York Medical School, University of York, York, United Kingdom, 6 Faculty of Medicine, Primary Care, Population Sciences and Medical Education, University of Southampton, Southampton, United Kingdom, 7 Bristol Dental School, University of Bristol, Bristol, United Kingdom

* charlotte.archer@bristol.ac.uk

**Data Availability Statement:** Data contains sensitive patient information so cannot be shared

## Abstract

### Purpose

Antidepressants are a first-line treatment for depression, yet many patients do not respond. There is a need to understand which patients have greater treatment response but there is little research on patient characteristics that moderate the effectiveness of antidepressants. This study examined potential moderators of response to antidepressant treatment.

### Methods

The PANDA trial investigated the clinical effectiveness of sertraline (n = 326) compared with placebo (n = 329) in primary care patients with depressive symptoms. We investigated 11 potential moderators of treatment effect (age, employment, suicidal ideation, marital status, financial difficulty, education, social support, family history of depression, life events, health and past antidepressant use). Using multiple linear regression, we investigated the appropriate interaction term for each of these potential moderators with treatment as allocated.

### Results

Family history of depression was the only variable with weak evidence of effect modification (p-value for interaction = 0.048), such that those with no family history of depression may have greater benefit from antidepressant treatment. We found no evidence of effect modification (p-value for interactions≥0.29) by any of the other ten variables.

publicly. Proposals for use of the data underlying the results presented in this study and requests for access should be directed to the data curator (fphs. pa@ucl.ac.uk) or the corresponding author (charlotte.archer@bristol.ac.uk). To gain access, researchers will need to sign a data access agreement with the PANDA study sponsor (University College London, London, UK; fphs. pa@ucl.ac.uk).

**Funding:** The PANDA trial is independent research commissioned by the National Institute for Health Research (NIHR) Programme Grant for Applied Research (RP-PG-0610-10048) [https://www.nihr. ac.uk/explore-nihr/funding-programmes/ programme-grants-for-applied-research.htm] awarded to GIL, NW, SG, TJP and RA.The views expressed in this publication are those of the author(s) and not necessarily those of the sponsor, UK National Health Service, NIHR, or UK Department of Health and Social Care. CA is funded on a Bristol, North Somerset and South Gloucestershire ICB launching fellowship (RCF 21/ 22-3.1) [https://bnssg.icb.nhs.uk/]. The funders had no role in study design, data collection and analysis, decision to publish, or preparation of the manuscript.

**Competing interests:** The authors have declared that no competing interests exist.

## Conclusion

Evidence for treatment moderators was extremely limited, supporting an approach of continuing discuss antidepressant treatment with all patients presenting with moderate to severe depressive symptoms.

## Background

Antidepressants are a very widely used treatment for depression [1]. General practitioners (GPs) usually prescribe selective serotonin reuptake inhibitors (SSRIs). However, it is estimated that 50% of patients do not respond to SSRI treatment [2]. It is unclear why a large proportion of patients do not improve after taking antidepressants. It is clinically important to know which patients are more likely to respond to antidepressant treatment, and which patients may be less likely to benefit. Much of the research published to date has focused on prognosis, independent of treatment. Marital status has consistently been found to be a prognostic indicator, with those who are either married or living with someone being more likely to have an improvement in their depressive symptoms during primary care treatment for depression [3–6]. Similarly, patients who have social support and who are better educated [3,7,8] are more likely to experience a reduction in their depressive symptoms. Likewise, those who consider themselves to have good physical health [9] and who have had previous 'adequate' antidepressant treatment [10] have been found to have a greater reduction in their depressive symptoms in response to pharmacological treatment. In contrast, adverse life events [11], financial difficulty [12,13], and a family history of depression [3,14] are associated with a worse prognosis.

However, whilst these variables may predict outcome independent of treatment, there is limited evidence on variables that are associated with a differential response to treatment. It is clinically useful to identify which patients will be more likely to have a better response to antidepressant medication. A recent study identified that older age, higher depression severity, higher neuroticism, less impairment in cognitive control and being employed were potential effect modifiers of antidepressant treatment, and were associated with better outcomes for patients randomised to receive sertraline compared to placebo [15]. In contrast, another study found no evidence that the severity or duration of depressive or anxiety symptoms moderated antidepressant response [16]. The latter study [16] was not funded by the pharmaceutical industry and did not restrict eligibility based on higher or lower thresholds of severity of symptoms, and therefore reflects the population currently receiving antidepressant treatment in the UK. An individual participant data (IPD) meta-analysis of seven placebo-controlled randomised trials found that, in addition to older age, greater scores on the Hamilton Depression Rating Scale (HRSD) subscales of guilt, anhedonia and insomnia, and the presence of suicidal ideation at baseline were potential moderators of antidepressant treatment [17].

We conducted secondary data analyses of PANDA trial [16] data to examine potential moderators of response to antidepressant treatment. We included variables that had previously been associated with differential response to antidepressant treatment (*a priori variables*)–age, employment status and suicidal ideation. In addition, we also investigated the extent to which variables that are prognostic of depression outcome more generally (marital status, financial difficulty, education, social support, life events, family history of depression, self-reported physical health and past antidepressant use) might also moderate the effectiveness of antidepressant treatment. We have previously reported that there was no evidence that treatment

response varied according to baseline severity or duration of symptoms [16] and hence these were not included in the list of variables explored in the present investigation.

## Methods

### Ethics statement

Ethical approval for the PANDA trial was obtained from the National Research Ethics Service Committee, East of England–Cambridge South (ref: 13/EE/0418). All participants provided written informed consent.

### Participants

This study is a secondary analysis of data collected as part of the PANDA trial, which was a multi-centre, placebo-controlled randomised trial to investigate the clinical effectiveness of sertraline in primary care (Lewis et al., 2019). Sertraline is one of the most widely used antidepressants in UK primary care, with the proportion of sertraline prescriptions increasing year on year [18].

Patients were eligible for the trial if they were aged between 18 to 74 years and there was uncertainty from the GP and patient about the possible benefit of an antidepressant. The PANDA research team used clinical uncertainty about the possible benefit of an antidepressant as a criterion for inclusion in order to avoid formal diagnostic or severity criteria, which are often not used by GPs in general practice, and are therefore more applicable to UK primary care. The exclusion criteria for PANDA were: antidepressant treatment in the past eight weeks; comorbid psychosis, schizophrenia, mania, hypomania, bipolar disorder, dementia, eating disorder, or major alcohol or substance abuse; and any medical contraindications for sertraline. These exclusion criteria were assessed by the patient's GP. A total of 655 patients were randomised to sertraline (n = 326) or placebo (n = 329), recruited from 179 GP surgeries between 26 Jan 2015 and Aug 31, 2017. Remote computer-generated code was used for randomisation, with researchers and patients masked to treatment allocation. The treatment allocation was stratified by recruitment site (Bristol, Liverpool, London, York), and by the severity and duration of patients' depressive symptoms which were both assessed by the Clinical Interview Schedule–Revised version (CIS-R) [19]. Imbalances in variables measured at baseline will not be adjusted for in the present analyses as they had little impact on the treatment effects reported in the main PANDA trial paper (Lewis et al., 2019). Due to attrition, there were 550 participants (266 in the sertraline group and 284 in the placebo group) with outcome data at six weeks; these formed the study data set for the present analyses and individuals were analysed in the groups to which they were randomised.

### Outcome

The primary outcome for the PANDA trial was the total score at 6 weeks on the Patient Health Questionnaire-9 (PHQ-9) [20]. To maximise power this was treated as a continuous variable.

### Potential moderators

The potential moderators were identified from the existing literature and were grouped into two categories: (1) *a priori* patient characteristics (that had previously been associated with differential response to antidepressant treatment); and (2) prognostic patient characteristics (that are prognostic of depression outcome more generally). Data on potential moderators were collected as part of a computerised self-report assessment conducted during the baseline visit. The assessments were conducted prior to randomisation. Further detail on data collection and

trial procedures were provided in the original trial paper [16]. As outlined below, where possible, we analysed quantitative measures as continuous variables to maximise statistical power. However, potential moderators were, for the purposes of clarity of interpretation, presented as categories.

**A priori characteristics.** Age was categorised into the following age bands: (1) 18–34 years old; (2) 35–54 years old; and (3) 55–74 years old. This is consistent with the categories reported in the main PANDA paper [16], with additional analyses undertaken for age as a continuous variable. For employment status, categories were collapsed into: (1) in paid employment *(employed full-time; employed part-time; on a government /employment training scheme);* and (2) not in paid employment *(studying at school, college or university; unemployed; permanently sick or disabled; looking after the home or family; retired)*. Responses (yes/no) on suicidal ideation were collapsed into: (1) no or low suicidal ideation *(scores 0–2)*; and (2) moderate or high suicidal ideation *(scores 3–4)*.

**Prognostic characteristics.** Marital status categories were: (1) married or living as married; (2) single; and (3) other *(separated; divorced; or widowed)*. The categories for financial difficulty were: (1): living comfortably or doing alright; (2) just about getting by; and (3) finding it difficult or very difficult to make ends meet.

Level of education was defined as the highest educational qualification, in the following categories: (1): A level or higher *(higher degree (e.g. M.A., PGCE) or equivalent; degree (e.g. B. Sc., B.A.) or equivalent; diploma (e.g. HND, NVQ, level 3) or equivalent; A-level or equivalent)*; (2) GCSE or other qualifications *(GCSE, O-level, CSE or equivalent; other qualifications)*; and (3) no formal qualifications. A-levels are UK qualifications that are usually taken at age 18, and are often required for entry to higher education, corresponding to 12th Grade in the US Education system. GCSEs are usually taken at age 16 at the end of UK compulsory education, corresponding to 10th Grade in the US.

Eight questions on social support from family or friends generated total scores between 1–24, and, were categorised into (1) low social support, (2) medium social support, and (3) high social support based on tertiles of the distribution of scores, with additional analyses undertaken for the underlying continuous variable. These questions are from the Health and Lifestyle Survey [21]. Eight questions (yes/no responses) on adverse life events (bereavement, separation or divorce, serious illness or injury, victim of crime, court appearances, debt, disputes with friends/relatives/neighbours and redundancy) were categorised as (1) no life events, (2) 1 or 2 life events, and (3) 3+ life events, again also analysed as a continuous variable.

Participants were also asked to give a 'yes' or 'no' response to: (1) if anyone in their immediate family has previously suffered from depression; (2) if their GP had previously prescribed anti-depressants for them; and (3) if they had a long-standing illness, disability or infirmity.

## Statistical analysis

The treatment effect in the PANDA trial was defined as the adjusted proportional difference in the PHQ-9 outcome score between the sertraline and placebo groups. This was obtained by exponentiating the regression coefficients from a linear regression of log-transformed PHQ-9 scores at six weeks, adjusted for the baseline PHQ-9 score and stratification (design) variables (baseline CIS-R score and depression duration) [16]. In the present analyses, for the log-transformed PHQ-9 scores, multiple linear regression techniques were used introducing the appropriate interaction term to test for effect modification by each potential moderator, using the likelihood ratio test and back-transforming the resultant coefficients to derive proportional 'differences'. Separate regression models were constructed for each moderator and were adjusted for the baseline PHQ-9 score and stratification (design) variables. This approach of

examining each moderator in a separate model was taken to aid the interpretation of results. We also constructed a single model including all potential moderator variables and their interaction terms.

## Results

### Baseline characteristics

Baseline characteristics of the sample were similar between treatment groups in terms of the stratification (design) variables (PHQ-9 total score, site, CIS-R depression duration, CIS-R total score and CIS-R depression severity score) and most of the potential treatment moderators (Table 1). There was a lower proportion of those who were married or living as married in the group randomised to treatment with sertraline (Table 1).

**Table 1. Comparison of baseline characteristics between randomisation groups for potential moderator variables.**

|  |  | Sertraline (n = 324) | Placebo (n = 329) |
|---|---|---|---|
| **Age (years; continuous): mean (SD)** |  | 39.7 (15.4) | 39.7 (14.6) |
| **Age (years; categories): n (%)** | 18–34 | 132 (40.7) | 134 (40.7) |
|  | 35–54 | 125 (38.6) | 134 (40.7) |
|  | 55–75 | 67 (20.6) | 61 (18.5) |
| **Marital status: n (%)** | Married or living as married | 116 (35.9) | 139 (42.3) |
|  | Single | 152 (47.0) | 144 (43.8) |
|  | Separated, divorced, or widowed | 55 (17.0) | 46 (14.0) |
| **Employment status: n (%)** | In paid employment | 209 (64.7) | 224 (68.1) |
|  | Not employed | 114 (35.3) | 105 (31.9) |
| Suicidal ideation[A]: n (%) | No or low suicidal ideation (0–2) | 191 (72.6) | 206 (75.7) |
|  | Moderate or high suicidal ideation (3–4) | 72 (27.4) | 66 (24.3) |
| **Financial difficulty: n (%)** | Living comfortably or doing alright | 180 (55.7) | 184 (55.9) |
|  | Just about getting by | 101 (31.2) | 103 (31.3) |
|  | Finding it difficult or very difficult | 42 (13.0) | 42 (12.8) |
| **Highest educational qualification: n (%)** | A Level or higher | 216 (66.8) | 234 (71.1) |
|  | GCSE, standard grade, or other | 92 (28.5) | 77 (23.4) |
|  | No formal qualification | 15 (4.6) | 18 (5.5) |
| **Social support score (continuous) (possible range 1–24) : mean (SD)** |  | 12.5 (4.0) | 12.8 (3.6) |
| **Social support score tertiles: n (%)** | Low (0–11) | 113 (35.0) | 98 (29.8) |
|  | Medium 12–15) | 95 (29.4) | 115 (35.0) |
|  | High (16+) | 115 (35.6) | 116 (35.3) |
| **Number of life events in past 6 months (continuous): mean (SD)** |  | 1.2 (1.2) | 1.2 (1.2) |
| **Number of life events in past 6 months (categories): n (%)** | 0 | 110 (34.1) | 106 (32.2) |
|  | 1–2 | 168 (52.0) | 178 (54.1) |
|  | 3+ | 45 (14.0) | 45 (13.7) |
| **Family history of depression: n (%)** | Yes | 205 (63.5) | 209 (63.5) |
|  | No | 118 (36.5) | 120 (36.5) |
| **Long standing illness: n (%)** | Yes | 140 (43.3) | 132 (40.1) |
|  | No | 183 (56.7) | 197 (59.9) |
| **Antidepressant in the past: n (%)** | Yes | 191 (59.1) | 200 (60.8) |
|  | No | 132 (40.9) | 129 (39.2) |

[A] n = 119 missing data.

## Treatment effect modification by potential moderators

The results from the linear regression models suggested that immediate family history of depression was the only variable with weak evidence of an interaction between treatment group and potential moderator (p-value for interaction: 0.048; Table 2). The adjusted proportional difference in PHQ-9 scores at 6 weeks for each stratum of the immediate family history of depression variable is presented in Table 3. There was no difference in depressive symptoms at 6 weeks in those given sertraline or placebo for those with a family history of depression (adjusted proportional difference in PHQ-9 score: 1.04 (95% CI: 0.91, 1.19); Table 3). Those with no family history of depression appeared to benefit from treatment (adjusted

**Table 2. P values from interaction tests for potential moderators from linear regression models of log-transformed PHQ-9 scores at 6 weeks.**

| Moderator | | n | P value for interaction test[A] |
|---|---|---|---|
| Age (years; continuous) | | 653 | 0.21 |
| Age (years; categories) | 18–34 | 266 | 0.22 |
| | 35–54 | 259 | |
| | 55–74 | 128 | |
| Marital status | Married or living as married | 255 | 0.81 |
| | Single | 296 | |
| | Separated, divorced, or widowed | 101 | |
| Employment status | In paid employment | 433 | 0.94 |
| | Not in paid employment | 219 | |
| Suicidal ideation | No or low suicidal ideation (0–2) | 397 | 0.12 |
| | Moderate or high suicidal ideation (3–4) | 138 | |
| Financial difficulty | Living comfortably or doing alright | 364 | 0.50 |
| | Just about getting by | 204 | |
| | Finding it difficult or very difficult | 84 | |
| Highest educational qualification | A Level or higher | 450 | 0.42 |
| | GCSE, standard grade, or other | 169 | |
| | No formal qualification | 33 | |
| Social support score (continuous) (possible range 1–24) | | 652 | 0.94 |
| Social support score tertiles | High (16+) | 231 | 0.26 |
| | Medium 12–15) | 210 | |
| | Low (0–11) | 211 | |
| Number of life events in past 6 months (continuous) | | 652 | 0.14 |
| Number of life events in past 6 months (categories) | 0 | 216 | 0.18 |
| | 1–2 | 346 | |
| | 3+ | 90 | |
| Family history of depression | No | 238 | 0.048 |
| | Yes | 414 | |
| Long standing illness | No | 272 | 0.41 |
| | Yes | 380 | |
| Antidepressant in the past | No | 261 | 0.68 |
| | Yes | 391 | |

[A] P values for likelihood ratio test of treatment effect modification from linear regression models comparing the model with the appropriate interaction term and the model without the appropriate interaction term but including the main effect of the moderator and the main effect of the treatment allocation. Each regression model was adjusted for the baseline PHQ-9 score and stratification (design) variables (PHQ-9 total score, site, CIS-R depression duration, CIS-R total score and CIS-R depression severity score).

**Table 3. Adjusted proportional difference in mean PHQ-9 scores at 6 weeks between randomisation groups for each stratum of family history.**

| | Baseline PHQ-9 | | | | 6 weeks PHQ-9 | | | | Adjusted proportional difference[A] | 95% CI |
|---|---|---|---|---|---|---|---|---|---|---|
| | Sertraline | | Placebo | | Sertraline | | Placebo | | | |
| | n | mean | n | mean | n | mean | n | mean | | |
| Family History of depression: No | 118 | 11.58 (5.90) | 119 | 11.87 (5.15) | 99 | 7.33 (5.57) | 103 | 8.83 (5.60) | 0.81 | (0.66, 0.99) |
| Family History of depression: Yes | 204 | 11.87 (5.86) | 209 | 12.39 (6.00) | 167 | 8.37 (5.65) | 181 | 8.72 (6.01) | 1.04 | (0.91, 1.19) |

[A] Values present adjusted proportional difference (treatment effect) following back-transformation of coefficients from regression models of log PHQ-9 scores for the sertraline group compared with the placebo group for each stratum of family history of depression. Values below one represent more desirable outcomes (i.e. greater treatment-derived benefit). Each regression model was adjusted for the baseline PHQ-9 score and stratification (design) variables (PHQ-9 total score, site, CIS-R depression duration, CIS-R total score and CIS-R depression severity score). Log transformed PHQ-9 scores = log (1+PHQ9 score at 6 weeks) in order to enable PHQ-9 scores of zero to be transformed.

proportional difference in PHQ-9 score: 0.81; Table 3). The interaction is illustrated graphically in Fig 1.

There was no evidence of an interaction for the other potential moderators investigated: age (in categories); employment status; suicidal ideation; marital status; financial difficulty; education; social support (in categories); life events (in categories); long standing illness; and past antidepressant use (all p-value for interactions >0.10, Table 2). No evidence of moderation was found when age, social support and life events were analysed as continuous variables (age, p = 0.21; social support, p = 0.94; life events, p = 0.18; Table 2).

There was no evidence of effect modification (all p values for interactions > 0.27) in the full model combining all potential moderator variables and their interaction terms, except for

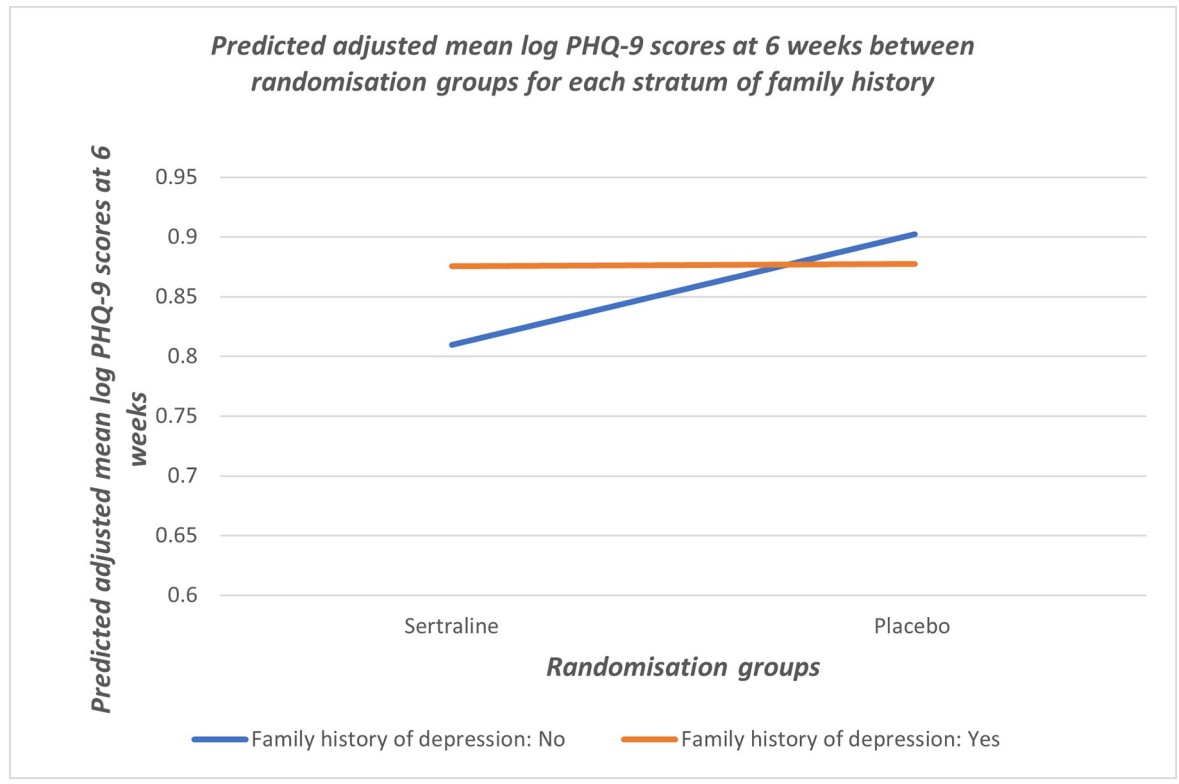

**Fig 1. Predicted adjusted mean log PHQ-9 scores at 6 weeks between randomisation groups for each stratum of family history.**

immediate family history of depression which was consistent with the results above–that is, there was weak evidence for effect modification (p = 0.05).

## Discussion

### Summary of main findings

Of the eleven variables that we investigated, the only variable with weak statistical evidence that it moderated the effect of sertraline compared with placebo on depressive symptoms at six weeks was immediate family history of depression. We found no evidence of effect modification by any of the other variables, including those previously identified as potential moderators (age, employment status and suicidal ideation), and those identified as other prognostic factors (marital status, financial difficulty, education, social support, life events, self-reported physical health and past antidepressant use).

### Strengths and limitations

The use of data from the PANDA trial enabled analysis of a sample size of 550 patients, which is larger than many other similar studies [22]. However, despite the large sample size, the PANDA trial was not designed to detect moderators of treatment effect and therefore is likely to be underpowered to estimate interaction effects robustly. Thus, the results of the present study should be interpreted with caution. In addition, given the multiple testing (11 tests of interaction) in this study, there is an increased likelihood of chance findings; in particular, while we ran interaction models for 11 different potential moderator variables, we only found weak evidence of interaction for one moderator (immediate family history of depression). Sub-group analysis have a useful role, generating hypotheses for future research. The variables reported in this manuscript were pre-defined based on prior evidence and prognostic literature and were the only variables that were tested for evidence of moderating treatment effects beyond the two reported in the original trial report [16].

In terms of the potential moderators investigated here, there may have been misclassification in exposure for some of these variables; asking participants to self-report if they have anyone in their immediate family with a history of depression may not be an accurate way of capturing this information. It is possible that some participants may not know, or may not remember, if their relatives have experienced depression. Further, it is not clear how participants who have relatives that have depressive symptoms, but have not sought or received a formal diagnosis of depression, have been captured in this study. Likewise, for marital status, participants in stable relationships could have been captured as either 'single' or 'living as married' depending on how they self-identified during the baseline assessment. Lastly, there may be other moderators of antidepressant treatment response that have not been measured or examined in this study, and we cannot rule out the possibility of confounding as well as false negatives for such effects.

### Comparison with previous literature

This study found weak evidence of an interaction between antidepressant response and immediate family history of depression. There was no difference in depressive symptoms at 6 weeks in those given sertraline or placebo for those with a family history of depression. However, those with no family history of depression appeared to benefit from treatment. The (point estimate of the) adjusted proportional difference for the sertraline group compared with the placebo group in those without a family history of depression was 19% on the PHQ-9, which is consistent with the findings from previous work that found that a 20% reduction in PHQ-9

scores represented a clinically important difference [23]–albeit that the upper confidence limit was very close to the null. Whilst previous studies have identified family history of depression as an indicator of worse prognosis independent of treatment [3,14], to our knowledge there was no prior evidence for family history of depression moderating antidepressant treatment response in the existing literature. It is possible that family history may be a proxy variable for genetic vulnerability, although research to investigate if antidepressant treatment response is influenced by genes has been inconclusive [24]. However, there is evidence that genetic markers may predict some of the individual differences in antidepressant response [25]. Conversely, other studies have found that family history of mental illness does not predict outcome of cognitive therapy and antidepressant treatment [26], or antidepressant treatment alone [27]. Therefore, there is a need for future research to examine whether the finding from the present study is robust and to understand the genetic and/or environmental mechanisms that may be involved.

When compared to existing literature [15,17], we did not find evidence of interactions between treatment response and age, employment status or suicidal ideation. The moderator identified in the meta-analysis by Noma, Furukawa [17] was baseline presence of suicidal ideation. In contrast, the results of the present study suggest that there was no statistical evidence of effect moderation in those with higher or lower levels of suicidal ideation. Webb, Trivedi [15] identified that older age and being in employment were associated with better outcomes for patients randomised to antidepressant treatment. In our study, the estimates for treatment effect did not differ between those who were in paid employment, and those who were not, and for each age group, with no statistical evidence of effect modification in both age and employment status.

## Clinical implications

Antidepressants are a first-line treatment option for those with moderate and severe depression, and it is clinically important to know which patients are more likely to respond to medication, and which patients may be less likely to benefit. However, the present study did not provide strong evidence of effect modification for any of the variables investigated. There are a number of factors that influence whether a clinician might recommend antidepressant treatment and shared decision making with the patient is essential. As none of the variables we examined help the clinician to decide on the likely effectiveness of antidepressants, we suggest that a discussion about taking antidepressants would be appropriate for all patients presenting with moderate and severe depressive symptoms.

## Research implications

There is a clear clinical need to identify which patients may be more likely to derive greater benefit from antidepressant treatment. This study highlights the importance of open access to data for building IPD datasets that will enable trial data to be combined to look at this question in the future and guide treatment decision making. Whilst the evidence for immediate family history moderating treatment response was weak, the finding warrants future research using IPD meta-analyses to examine whether this finding can be replicated, and to investigate any genetic or environmental mechanisms that may be involved.

## Conclusions

Findings from this study support an approach of continuing to offer antidepressant treatment to patients who present with moderate to severe depressive symptoms. However, there is a

need for more work in this area, particularly to understand if the weak evidence for family history of depression moderating antidepressant treatment response can be replicated.

## Acknowledgments

### Declarations

We are grateful to all the patients and GP surgery staff who took part in this research, and the support provided by the local clinical research networks (CRNs) and the University College London Hospitals Biomedical Research Centre Mental Health theme. This study was also supported by the National Institute for Health and Care Research Bristol Biomedical Research Centre. We also thank colleagues who contributed to the study through recruitment, administrative help, and other advice. In particular, Stephanie J MacNeill, for providing statistical advice.

## Author Contributions

**Conceptualization:** Charlotte Archer, David Kessler, Gemma Lewis, Ricardo Araya, Larisa Duffy, Simon Gilbody, Glyn Lewis, Tony Kendrick, Tim J. Peters, Nicola Wiles.

**Formal analysis:** Charlotte Archer, Gemma Lewis, Tim J. Peters.

**Funding acquisition:** Nicola Wiles.

**Methodology:** Charlotte Archer, David Kessler, Gemma Lewis, Ricardo Araya, Larisa Duffy, Simon Gilbody, Glyn Lewis, Tony Kendrick, Tim J. Peters, Nicola Wiles.

**Supervision:** David Kessler, Nicola Wiles.

**Writing – original draft:** Charlotte Archer.

**Writing – review & editing:** David Kessler, Gemma Lewis, Ricardo Araya, Larisa Duffy, Simon Gilbody, Glyn Lewis, Tony Kendrick, Tim J. Peters, Nicola Wiles.

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
