## [Decision Letter · Decision Letter 0]

29 Nov 2023

PONE-D-23-29027What predicts response to antidepressants for people with depression in primary care? An analysis of moderators in the PANDA trialPLOS ONE

Dear Dr. Archer,

Thank you for submitting your manuscript to PLOS ONE. After careful consideration, we feel that it has merit but does not fully meet PLOS ONE’s publication criteria as it currently stands. Therefore, we invite you to submit a revised version of the manuscript that addresses the points raised during the review process.

We look forward to receiving your revised manuscript.

Kind regards,

Dickens Akena, Ph.D

Academic Editor

PLOS ONE

Reviewers' comments:

Reviewer's Responses to Questions

**Comments to the Author**

1. Is the manuscript technically sound, and do the data support the conclusions?

Reviewer #1: Partly

Reviewer #2: Partly

Reviewer #3: Yes

Reviewer #4: Yes

2. Has the statistical analysis been performed appropriately and rigorously? 

Reviewer #1: I Don't Know

Reviewer #2: I Don't Know

Reviewer #3: I Don't Know

Reviewer #4: Yes

3. Have the authors made all data underlying the findings in their manuscript fully available?

Reviewer #1: Yes

Reviewer #2: Yes

Reviewer #3: Yes

Reviewer #4: No

4. Is the manuscript presented in an intelligible fashion and written in standard English?

Reviewer #1: Yes

Reviewer #2: No

Reviewer #3: Yes

Reviewer #4: Yes

5. Review Comments to the Author

Reviewer #1: Important note: This review pertains only to ‘statistical aspects’ of the study and so ‘clinical aspects’ [like medical importance, relevance of the study, ‘clinical significance and implication(s)’ of the whole study, etc.] are to be evaluated [should be assessed] separately/independently. Further please note that any ‘statistical review’ is generally done under the assumption that (such) study specific methodological [as well as execution] issues are perfectly taken care of by the investigator(s). This review is not an exception to that and so does not cover clinical aspects {however, seldom comments are made only if those issues are intimately / scientifically related & intermingle with ‘statistical aspects’ of the study}. Agreed that ‘statistical methods’ are used as just tools here, however, they are vital part of methodology [and so should be given due importance]. I look at the manuscript in/with statistical view point, other reviewer(s) look(s) at it with different angle so that in totality the review is very comprehensive. However, there should be efforts from authors side to improve (may be by taking clues from reviewer’s comments). Therefore, please do not limit the revision only (with respect) to comments made here.

COMMENTS: There are a few serious issues about which I have different opinion and such observations/concerns are given below:

Firstly, the ‘Title’ may mention that it is a secondary data analysis [nothing wrong in secondary data analysis, only it should be clear to potential readers, I guess]. This is (anyway) mentioned later on page 5 [We conducted secondary data analyses of PANDA trial [17] data to examine potential moderators of response to antidepressant treatment] anyway.

Secondly, one very basic/fundamental question: In the back drop of the fact that the original trial article [Lewis G, Duffy L, Ades A, Amos R, Araya R, Brabyn S, et al. The clinical effectiveness of sertraline in primary care and the role of depression severity and duration (PANDA): a pragmatic, double-blind, placebo-controlled randomised trial. The Lancet Psychiatry. 2019;6(11)] clearly concludes “the main benefits in the first 6 weeks of treatment with sertraline are on reduction of anxiety symptoms, such as worry and restlessness, rather than an improvement in depressive symptoms” and also says that “Our finding that sertraline was not effective for depressive symptoms at 6 weeks is inconsistent with previous studies”, I wonder if this database is expected to through light on problem/question in hand [“What predicts response to antidepressants for people with depression in primary care?”] for the very purpose of this small ‘paper exercise’ [[i.e., no design or execution or data collection]? In ‘Strengths and limitations’ section, though it is stated that “the PANDA trial was not designed to detect moderators of treatment effect and therefore is likely to be underpowered to estimate interaction effects robustly. Thus, the results of the present study should be interpreted with caution” however, one can not do away just by stating/adding such cautionary note, in my opinion.

Next, refer to ‘Methods-Outcome’ section where it is stated that “primary outcome for PANDA was the total score at 6 weeks on the Patient Health Questionnaire-9 (PHQ-9) and to maximise power this was treated as a continuous variable” which is totally “DISAGREED” because it is very erroneous. This is not the correct way to maximise/increasing/enhancing the power. Statement (on face value) that ‘treating any non-continuous variable as a continuous variable maximises (increase) the power’ is not only surprising but [such an erroneous statement] is not expected from present learned authors’ team. If you have something else in mind, please clarify. This is my first-hand {natural & immediate} reaction on the statement as it appears [& as I have understood it]. Please remember that this is a scientific/academic document and so all details should be clearly/correctly communicated (do not take readers’ for granted). In next section [Potential moderators] you said “In many cases the potential moderators were for the purposes of clarity of interpretation and presentation considered as categories; however, to maximise power many of the quantitative(?) measures were also analysed as continuous variables.” Is that correct? Surprising/wonderful.

Mind you that the PHQ‐9 (used here, though correctly, as a brief depression severity measure) is likely to yield data that are in ‘ordinal’ level of measurement [and not in ratio level of measurement for sure {as the score two times higher does not indicate presence of that parameter/phenomenon as double}]. Then application of suitable non-parametric (or distribution free) test(s) is/are indicated/advisable [even if distribution may be ‘Gaussian’ (also called ‘normal’)]. May check (level of measurement) for other scores (example: CIS-R depression severity score, social support score).

There is clear mention about ‘allocation’ {that Patients were randomly assigned (1:1) with a remote computer-generated code to sertraline or placebo, and were stratified by severity, duration, and site with random block length} in original trial article, but in present article (expected as this is an independent article and not one in series) there no such indication. More discussion on ‘sample size’ is also expected. Only saying that “The use of data from the PANDA trial enabled analysis of a sample size of 550 patients, which is larger than many other similar studies” is not at all sufficient, in my opinion.

As pointed out in ‘important note’ above “This review pertains only to ‘statistical aspects’ of the study and so ‘clinical aspects’ should be assessed separately/independently [one should carefully consider/look at the clinical implications of the study]. In my opinion, to make this article acceptable (which is quite possible and easy), a small amount of re-vision (re-drafting) may be needed. The respected ‘Editor’ may consider accepting/further processing as otherwise the draft is nicely worded. ‘Major revision’ is recommended, assuming that the respected editor would like to give chance to authors for improvement of the manuscript.

Reviewer #2: 1. The quality of the work is lacking. Use of different fonts, unjustified work, no line numbers; all these make the work tedious to review and give feedback

2. The title and the entire paper indicates that the authors tested various antidepressants yet all they assessed was sertraline. This is very misleading.

3. In the background and abstract the authors make a statement that antidepressants are the first line treatment for depression which is not true. First line treatment for depression is psychotherapy. Antidepressants are first line for severe depression

4. The conclusion in both the abstract and the main document need to be phrased better. The conclusion drawn from this work is rather erroneous given that the team only assessed on antidepressant

5. Under the section of participants; lines 4 and 5 are confusing. what was the uncertainty about..the benefits of antidepressant or diagnosis of depression

6.The authors interchangeably use the terms depressive symptoms and depression yet they assessed for depressive symptoms not depression.

7. Was the exclusion criteria by self report or by assessment tools. this needs to be explained properly.

8. In the out comes, the authors now introduce measurements by PHQ-9 at 6 weeks instead of using the CIS-R used at baseline. And again later on in the statistical analysis they are comparing baseline and 6/52 PHQ-9 scores. there was no mention of using the PHQ-9 at baseline in the methods. what was the rationale for measures at 6 weeks why not 3 or 6 months

9. Page 8, what tool was used to measure social support. If none, how did you come up with the scores.

10. please include a section of data collection tools and study procedure in your methods section.

11. The entire paper but mostly the methods sections needs to be written better

12. In the results section, the authors indicate that the baseline characteristics were similar for the two groups yet no p-values are provided in table 1 to this effect

13. table 1, there is no need to add % in all the cells. simply put it as (%) in the top cell of each column

14. The randomization process needs to be explained better. who conducted it, was there blinding or not,etc

Reviewer #3: The authors have conducted a secondary analysis on data from the PANDA trial to examine the potential mediators for response to antidepressant treatment. This is an important topic as knowledge of these mediators can help clinicians to make decisions on who would most likely benefit from antidepressant treatment. The paper is generally well presented.

My first comment is about table 1 which presents the baseline characteristics of the participants in both arms. The authors report that there was no statistical difference in the baseline characteristics of the two arms. I want to believe that a statistical test was done for this. Can the findings of this test also be presented in table 1 for clarity?

Table 1 has a footnote A where n=119 (for suicidal ideation). However, from the table the n for suicide ideation seems to be 535. Can you clarify on this or correct it?

Secondly, in the discussion section where the authors are comparing their findings with findings from other studies, they report that those aged 18-35 had greater treatment derived benefit than those aged 55-74 and that that individuals with higher levels of suicidal ideation derived less treatment benefit. However, these findings are not mentioned anywhere in the results. It leaves the readers wondering where these results came from. You may want to include them in your results section (text or tables).

Reviewer #4: GENERAL:

An important area of work to provide for precision medicine, with patient centered medical care for mental wellness.

SPECIFIC:

1. The background attemtps to set the scene and this is good. Still, it is based on evidence synthesis and single studies. This creates confusion especially the flow from single studies, to a meta-analysis to single studies again. It would be key that the progonstic factors come from syntheses only. So that the authors could provide a synthesis of the recent studies done after the meta-analysis.

2.Ethical considerations: Was the observational study of the PANDA trial approved by the ethical committees? Or was this covered in the initial approval?

3.Table 1: Suggests randomization did not achieve a balance for those who were categorised as married, 36% vs 42%, intervention vs placebo. How did this affect the adjusted analysis?

4.The description of weak evidence for interaction based on the p-value may not be appropriate. The p-value is <0.05 (<0.29) cut off suggests that the relationship between the exposure/interactor and outcome is statisticaly significant. The authors acknolwegde that the PANDA trial "...was not designed to detect moderators of treatment effect and therefore is likely to be underpowered to estimate interaction effects robustly..." which may explain the "...weak evidence..." that could better be described in form of statistical significance.

5. It would be desireable to present the findings mostly in systematc reviews rather than single studies. Could the authors check for systematic reviews about the same and place their findings in this eveidence base?

6. The authors should consider summing the eunanswered research questions under "research implications". In this regard, are there any implications for relevant policy guidelines?

6. PLOS authors have the option to publish the peer review history of their article (what does this mean?). If published, this will include your full peer review and any attached files.

Reviewer #1: No

Reviewer #2: No

Reviewer #3: No

Reviewer #4: **Yes: **Ekwaro A OBUKU

---

## [Author Response · Author response to Decision Letter 0]

25 Jan 2024

Editors’ comments 

The figure file names have been updated. 

The funding information section has been updated to match. 

3. If there are ethical or legal restrictions on sharing a de-identified data set, please explain them in detail (e.g., data contain potentially identifying or sensitive patient information) and who has imposed them (e.g., an ethics committee). Please also provide contact information for a data access committee, ethics committee, or other institutional body to which data requests may be sent.

The data availability statement has been updated to: 

Data contains sensitive patient information so cannot be shared publicly. Proposals for use of the data underlying the results presented in this study and requests for access should be directed to glyn.lewis@ucl.ac.uk. To gain access, researchers will need to sign a data access agreement with the PANDA study sponsor (University College London, London, UK).

Comments to the Author

1. Is the manuscript technically sound, and do the data support the conclusions?

Reviewer #1: Partly

Reviewer #2: Partly

Reviewer #3: Yes

Reviewer #4: Yes

2. Has the statistical analysis been performed appropriately and rigorously?

Reviewer #1: I Don't Know

Reviewer #2: I Don't Know

Reviewer #3: I Don't Know

Reviewer #4: Yes

3. Have the authors made all data underlying the findings in their manuscript fully available?

Reviewer #1: Yes

Reviewer #2: Yes

Reviewer #3: Yes

Reviewer #4: No

4. Is the manuscript presented in an intelligible fashion and written in standard English?

Reviewer #1: Yes

Reviewer #2: No

Reviewer #3: Yes

Reviewer #4: Yes

Reviewer #1: 

Important note: This review pertains only to ‘statistical aspects’ of the study and so ‘clinical aspects’ [like medical importance, relevance of the study, ‘clinical significance and implication(s)’ of the whole study, etc.] are to be evaluated [should be assessed] separately/independently. Further please note that any ‘statistical review’ is generally done under the assumption that (such) study specific methodological [as well as execution] issues are perfectly taken care of by the investigator(s). This review is not an exception to that and so does not cover clinical aspects {however, seldom comments are made only if those issues are intimately / scientifically related & intermingle with ‘statistical aspects’ of the study}. Agreed that ‘statistical methods’ are used as just tools here, however, they are vital part of methodology [and so should be given due importance]. I look at the manuscript in/with statistical view point, other reviewer(s) look(s) at it with different angle so that in totality the review is very comprehensive. However, there should be efforts from authors side to improve (may be by taking clues from reviewer’s comments). Therefore, please do not limit the revision only (with respect) to comments made here.

4. Firstly, the ‘Title’ may mention that it is a secondary data analysis [nothing wrong in secondary data analysis, only it should be clear to potential readers, I guess]. This is (anyway) mentioned later on page 5 [We conducted secondary data analyses of PANDA trial [17] data to examine potential moderators of response to antidepressant treatment] anyway.

The title has been revised to include this. 

5. Secondly, one very basic/fundamental question: In the back drop of the fact that the original trial article [Lewis G, Duffy L, Ades A, Amos R, Araya R, Brabyn S, et al. The clinical effectiveness of sertraline in primary care and the role of depression severity and duration (PANDA): a pragmatic, double-blind, placebo-controlled randomised trial. The Lancet Psychiatry. 2019;6(11)] clearly concludes “the main benefits in the first 6 weeks of treatment with sertraline are on reduction of anxiety symptoms, such as worry and restlessness, rather than an improvement in depressive symptoms” and also says that “Our finding that sertraline was not effective for depressive symptoms at 6 weeks is inconsistent with previous studies”, I wonder if this database is expected to through light on problem/question in hand [“What predicts response to antidepressants for people with depression in primary care?”] for the very purpose of this small ‘paper exercise’ [[i.e., no design or execution or data collection]? In ‘Strengths and limitations’ section, though it is stated that “the PANDA trial was not designed to detect moderators of treatment effect and therefore is likely to be underpowered to estimate interaction effects robustly. Thus, the results of the present study should be interpreted with caution” however, one can not do away just by stating/adding such cautionary note, in my opinion.

While PANDA was not designed to examine moderators of treatment response, it is still one of the largest trial datasets examining the effectiveness of an antidepressant. In the limitations section (page 16, paragraph) we have included a statement making clear that it is important that the results of the study are interpreted with caution because of this. In the manuscript, we present the strength of evidence from our formal tests of interaction (p value for interaction between treatment allocation and each potential moderator) in Table 1. As others have highlighted (Dziak, Dierker & Abar, 2018. The interpretation of statistical power after the data have been gathered. Current Psychology, 39, 870-877), a post-hoc power calculation would not be appropriate but exploratory analyses (interpreted with appropriate caution) are important for increasing knowledge (added to limitations section, page 16, paragraph 1). 

6. Next, refer to ‘Methods-Outcome’ section where it is stated that “primary outcome for PANDA was the total score at 6 weeks on the Patient Health Questionnaire-9 (PHQ-9) and to maximise power this was treated as a continuous variable” which is totally “DISAGREED” because it is very erroneous. This is not the correct way to maximise/increasing/enhancing the power. Statement (on face value) that ‘treating any non-continuous variable as a continuous variable maximises (increase) the power’ is not only surprising but [such an erroneous statement] is not expected from present learned authors’ team. If you have something else in mind, please clarify. This is my first-hand {natural & immediate} reaction on the statement as it appears [& as I have understood it]. Please remember that this is a scientific/academic document and so all details should be clearly/correctly communicated (do not take readers’ for granted). 

The authors (Kroenke & Spitzer, 2001) of the PHQ-9 state the instrument can be used as a continuous variable, and many similar studies have used it in this way (e.g Saether et al, 2022. Moderators of treatment effect of Prompt Mental Health Care compared to treatment as usual: Results from a randomized controlled trial. Behaviour and research therapy, vol 158). There is consensus that depressive symptoms are best described as a continuum, so we are not treating a binary variable as continuous. Methodological literature has shown that categorizing continuous variables without strong justification likely results in reduced statistical power (MacCallum et al., 2002; Fedoro et al., 2009; DeCoster et al., 2009; Shentu & Xie, 2010).

7. In next section [Potential moderators] you said “In many cases the potential moderators were for the purposes of clarity of interpretation and presentation considered as categories; however, to maximise power many of the quantitative(?) measures were also analysed as continuous variables.” Is that correct? Surprising/wonderful.

We have reworded this section (‘potential moderators’, page 7, paragraph 3) to make it clearer for the reader. 

8. Mind you that the PHQ‐9 (used here, though correctly, as a brief depression severity measure) is likely to yield data that are in ‘ordinal’ level of measurement [and not in ratio level of measurement for sure {as the score two times higher does not indicate presence of that parameter/phenomenon as double}]. Then application of suitable non-parametric (or distribution free) test(s) is/are indicated/advisable [even if distribution may be ‘Gaussian’ (also called ‘normal’)]. May check (level of measurement) for other scores (example: CIS-R depression severity score, social support score).

As noted in the response to point 6, the authors (Kroenke & Spitzer, 2001) of the PHQ-9 state the instrument can be considered as a continuous variable (and is often used in applied health research in this way). This is in keeping with how the original PANDA trial analysed this data, and how other similar studies have done (e.g Saether et al, 2022. Moderators of treatment effect of Prompt Mental Health Care compared to treatment as usual: Results from a randomized controlled trial. Behaviour and research therapy, vol 158)

9. There is clear mention about ‘allocation’ {that Patients were randomly assigned (1:1) with a remote computer-generated code to sertraline or placebo, and were stratified by severity, duration, and site with random block length} in original trial article, but in present article (expected as this is an independent article and not one in series) there no such indication. More discussion on ‘sample size’ is also expected. Only saying that “The use of data from the PANDA trial enabled analysis of a sample size of 550 patients, which is larger than many other similar studies” is not at all sufficient, in my opinion.

We have added in some more information about the randomisation process to page 6, paragraph 3. We note in the limitations section, using appropriately cautious language, that “despite the large sample size, the PANDA trial was not designed to detect moderators of treatment effect and therefore is likely to be underpowered to estimate interaction effects robustly. Thus, the results of the present study should be interpreted with caution”. As outlined in our response to point 5, a post-hoc power calculation would not be appropriate.

As pointed out in ‘important note’ above “This review pertains only to ‘statistical aspects’ of the study and so ‘clinical aspects’ should be assessed separately/independently [one should carefully consider/look at the clinical implications of the study]. In my opinion, to make this article acceptable (which is quite possible and easy), a small amount of re-vision (re-drafting) may be needed. The respected ‘Editor’ may consider accepting/further processing as otherwise the draft is nicely worded. ‘Major revision’ is recommended, assuming that the respected editor would like to give chance to authors for improvement of the manuscript.

Thank you for your review, and we hope that the edits that we have made have addressed your points. 

Reviewer #2: 

10. The quality of the work is lacking. Use of different fonts, unjustified work, no line numbers; all these make the work tedious to review and give feedback

We were surprised that reviewer 2 raised issues with the quality of our manuscript. . The other reviewers noted the importance of the work, and stated that it was well-presented. We hope that the revisions we have made have improved the clarity of article for reviewer 2. We have added line numbers. All of the formatting (fonts etc) is in line with guidance issued by Plos One. 

11. The title and the entire paper indicates that the authors tested various antidepressants yet all they assessed was sertraline. This is very misleading.

We have changed the title to reflect this. We have also added a line to the methods section (page 6, paragraph 2) to explain that sertraline is one of the most commonly prescribed antidepressants in UK primary care. 

12. In the background and abstract the authors make a statement that antidepressants are the first line treatment for depression which is not true. First line treatment for depression is psychotherapy. Antidepressants are first line for severe depression

We have reworded this section (page 4, paragraph 1) to make it clear that antidepressants are a widely used treatment for depression.

13. The conclusion in both the abstract and the main document need to be phrased better. The conclusion drawn from this work is rather erroneous given that the team only assessed on antidepressant

As highlighted above, sertraline is one of the most commonly used antidepressants in UK primary care. We have changed the title to make clear that sertraline was the specific antidepressant used in the PANDA study but feel it is appropriate to retain reference to antidepressants in drawing conclusions from the study. The other three reviewers had not requested any changes to our conclusions. 

14. Under the section of participants; lines 4 and 5 are confusing. what was the uncertainty about the benefits of antidepressant or diagnosis of depression

We have clarified this in the manuscript. 

15. The authors interchangeably use the terms depressive symptoms and depression yet they assessed for depressive symptoms not depression.

We have used the term ‘depressive symptoms’ where appropriate to capture the PANDA trial population. Where we refer ‘depression’ it is mostly in relation the variable ‘family history of depression’. In the limitations section (page 16, paragraph 1), we note that “asking participants to self-report if they have anyone in their immediate family with a history of depression may not be an accurate way of capturing this information. It is possible that some participants may not know, or may not remember, if their relatives have experienced depression. Further, it is not clear how participants who have relatives that have depressive symptoms, but have not sought or received a formal diagnosis of depression, have been captured in this study”. We have changed ‘depression’ to ‘depressive symptoms’ in the clinical implications section (page 18, paragraph 2).

16. Was the exclusion criteria by self report or by assessment tools. this needs to be explained properly.

We have clarified this in the manuscript (page 6, paragraph 3). 

17. In the out comes, the authors now introduce measurements by PHQ-9 at 6 weeks instead of using the CIS-R used at baseline. And again later on in the statistical analysis they are comparing baseline and 6/52 PHQ-9 score

---

## [Editor Report · Decision Letter 1]

27 Feb 2024

What predicts response to sertraline for people with depression in primary care? A secondary data analysis of moderators in the PANDA trial

PONE-D-23-29027R1

Dear Dr. Archer

We’re pleased to inform you that your manuscript has been judged scientifically suitable for publication and will be formally accepted for publication once it meets all outstanding technical requirements.

Kind regards,

Dickens Akena, Ph.D

Academic Editor

PLOS ONE
---

## [Editor Report · Acceptance letter]

29 Apr 2024

PONE-D-23-29027R1 

PLOS ONE

Dear Dr. Archer, 

I'm pleased to inform you that your manuscript has been deemed suitable for publication in PLOS ONE. Congratulations! Your manuscript is now being handed over to our production team.

Kind regards, 

on behalf of

Dr. Dickens Akena 

Academic Editor

PLOS ONE